# The Sex Ratio Indicates the Conclusion and Onset of Population Cycles in the Beet Webworm *Loxostege sticticalis* L. (Lepidoptera: Pyralidae)

**DOI:** 10.3390/insects14100781

**Published:** 2023-09-25

**Authors:** Yunxia Cheng, Min Hu, Aiguo Kang, Yonghong Xiao, Lizhi Luo, Xingfu Jiang

**Affiliations:** 1State Key Laboratory for Biology of Plant Diseases and Insect Pests, Institute of Plant Protection Chinese Academy of Agricultural Sciences, Beijing 100193, China; yxcheng@ippcaas.cn (Y.C.); 15729915853@163.com (M.H.); lzluo@ippcaas.cn (L.L.); 2Plant Protection and Inspection Station of Kangbao County, Zhangjiakou 076650, China; kangaiguo998@163.com; 3College of Life Sciences, Jinggangshan University, Ji’an 343009, China; yonghongxiao01@126.com

**Keywords:** beet webworm, *Loxostege sticticalis*, sex ratio, mating frequency, population dynamics

## Abstract

**Simple Summary:**

The beet webworm *Loxostege sticticalis* L. (Lepidoptera: Pyralidae) is one of the most disastrous pest insects that outbreak periodically in northern China. Developing a definite indicator depending on defining the ending and beginning of the occurrence period cycle is urgent for the population forecast and theoretical study of the beet webworm. Looking through the 38-year historical population survey data, we found a close connection between the sex ratio and the population occurrence cycle. To clarify how the sex ratio connects with the population occurrence cycle, we measured the maximum mating potential of both females and males and the population's net reproductive rate under different sex ratios. Our results showed that the population net reproductive rate fits a parabolic curve analysis according to the variation of sex ratio and topped at 1.82, which indicates that the population may begin a new period of the cycle when the sex ratio achieves 1.82. In contrast, collapse may happen when the sex ratio is less than one. Our findings accurately forecast long-term dimension population occurrence dynamics in the beet webworm.

**Abstract:**

Beet webworms, *Loxostege sticticalis* L. (Lepidoptera: Pyralidae), are one of the most destructive pest insects in northern China, and their populations outbreak periodically. Developing an indicator that defines the ending and beginning of the occurrence period cycle is urgent for the population forecast and theoretical study. The sex ratio can be a primary pathway through which species regulate population size. We measured the maximum mating potential of both females and males and the population net reproductive rate under different sex ratios (e.g., 3:1, 2:1, 1:1, 1:2, 1:3). The maximum mating frequency of males was 2.91 times that of females. The progeny contribution per mating decreased with increased mating times in males. The variation in population net reproductive rate affected by the sex ratio fits the parabolic curve analysis and peaked at 1.82 for females vs. males. Our results illustrate the quantitative connection phenomenon shown by the historical data: population outbreaks occur at a sex ratio of two or more and collapse at a sex rate lower than one. Simultaneously, the sex ratio may be utilized as a definite indicator for the beginning and end of the future occurrence cycle in the beet webworm.

## 1. Introduction

The sex ratio is the numerical sex allocation in a population [1]. For the species, there are two kinds of sexually determined homologous gametes; the sex ratio should be 1:1 hereditarily [2,3,4]. While this value varies depending on the physical or external factors encountered during the development of individuals [5,6,7], the sex ratio is one of the most critical biological characteristics influencing the population development of arthropods. Such development control occurs by increasing or decreasing the chance of mating [8,9,10]. This happens especially for migratory insect species, where population outbreak years are often at a certain sex ratio [11]. This indicates that the sex ratio as well as migratory behavior may intensify population occurrence [12]. The sex ratio is far more easily tested than other factors that may simultaneously contribute to population outbreaks, such as natural enemies, population quantity, and host plants [13,14,15]. If the sex ratio can be verified as an authentic indicator of a population outbreak, the population outbreak disaster forecast for migratory insect species would also become more manageable. The sex ratio and the factors that contribute to it are now remarkably well understood [16,17,18], and it is even known that sex allocation affects population dynamics [19,20]. However, the application of the sex ratio in the evaluation of population dynamics is minimal, not to mention the explicit value of the sex ratio that contributes to population outbreaks.

In a population with a male-biased sex ratio, individuals may divert more investment into competition for mating, consequently decreasing their investment in reproduction [21]. The female-biased sex ratio is thought to facilitate population growth [22]. Especially in polyandrous populations, multiple matings are available to release potential reproductive resources [23,24,25,26,27]. However, the maximum mating frequency often does not equal the maximum reproductive resources released. This is because the valid reproductive resource is often assumed to be the number of progenies, and the maximum mating frequency of females and males may hinder achieving the maximum number of progenies [28,29]. Hence, there would be an equivalent between the mating frequency and the quantity of progeny reproduced, which would be achieved by the sex ratio in the natural habitat. Theoretically, the optimal sex ratio should be that under which the maximum number of offspring is facilitated [30]. Although adult senescence trades off with reproduction in many insects [22], this study assesses the optimal sex ratio under which the maximum number of offspring can be obtained, but not later senescence.

The beet webworm, *Loxostege sticticalis* L. (Lepidoptera: Pyralidae), is a migratory and destructive pest of crops and fodder plants and weeds in the northern temperate zone from 36° N to 55° N [31,32,33]. These plants include soybeans, potatoes, corn, peanuts, sunflowers, sugar beets, alfafa, and so on. To date, there have been over 50 families and over 300 plant species reported damaged by larval beet webworms. The larvae, on population outbreak, can eat up plant leaves, stems, and even the whole field of plants, causing serious destruction or extinction to production [32].

Every 20 years or so, population outbreaks of this species occur and last for years to tens of years [34]. Periodic population outbreaks threaten agricultural production and ecological stability in northern China [35]. The annual migration of individuals from the overwintering areas of the species to crop production regions in northern China is one of the most critical factors inducing population outbreaks [12]. Population outbreaks were closely related to the adult sex ratio, determined by the number of females and males monitored by light traps at each monitoring point in China. The extent of the occurrence of the first generation of larvae can be evaluated according to the number of overwintering generations of adult beet webworms trapped daily and coped with by the field survey and dissection of ovaries. Unfortunately, no study has been published on the connection between the sex ratio and population dynamics in beet webworms.

Our study analyzed 38 years of historical data from 1979 to 2016 on monitoring adults trapped under light traps (Appendix A, Figure A1). First, the generation larval damage area in Kangbao County found that the sex ratio was not stabilized at the ideal value of 1:1 as a typical Lepidoptera species. Nevertheless, it varied from 0.7 to 2.2 (the number of females vs. the number of males), and the first generation of larval damage area was strongly positively correlated with the sex ratio in this range. In addition, during the damage period of beet webworms, when the sex ratio was lower than one, regardless of the overwinter adult number, the first-generation larval damage area would be very low. This is called population collapse and indicates the end of a fluctuation period. For example, the endings in 1986 and 2009, separately (Appendix A, Figure A1). Later, the population enters an interoccurrence period, during which the larval number and larval damage area are shallow, which lasts for years (e.g., 1987–1996, 2010–2016). This period would not end until the sex ratio recovered from a bottom point of less than one to the next nearest top point of more than two (e.g., 1997). Then, the population entered another occurrence period (e.g., 1997–2009). During a new occurrence period, the sex ratio fluctuated from one to two. This period usually continued for 10–12 years until the value decreased to another low point lower than one and recovered again. This process repeated itself, and the beet webworm population broke out periodically.

This finding leads to our thinking: since a female-biased sex ratio promotes population prosperity and a male-biased sex ratio leads to the decline of the population, why is it the sex ratio of two that causes the outbreak of population in the beet webworm? We inferred that there must be an ideal sex ratio between females and males where the population’s net reproductive rate can reach its maximum. The ideal sex ratio may be based on the capacity for mating or reproduction, including the mating frequency of each sex or the potential of offspring that each sex can contribute to the population. Since there has been no report on the sex ratio applied in the forecast of the beet webworm population, the ideal sex ratio verified here may bring new interest to the forecast of beet webworm population outbreaks.

## 2. Materials and Methods

### 2.1. Insect Rearing

The beet webworms tested in this study were from a laboratory population that was established in 1980. Larvae were reared on lamb’s quarters, *Chenopodium album* L. (Chenopodiaceae), at 22 °C, 75% relative humidity, and a 16:8 (L:D) photoperiod. These individuals were kept in a jar (650 mL) at a density of 30 larvae per jar until they matured. The mature larvae were transferred to a new jar containing 10% water from sterilized soil for pupating. Adults were given a 10% glucose solution via cotton wicks on the cage lid as food.

### 2.2. Testing the Mating Potential of Male Beet Webworms

Mating potential was assessed by the maximum mating frequency. Newly emerged female and male moths were paired in 245-mL cylinders constructed with transparent plexiglass, and a 10% glucose solution was provided via cotton wicks on the cage lid as food. A red flashlight was used to observe the mating behavior of the beet webworm every 20 min in the last 4 h of the dark period. Both mating frequency and mating duration were recorded during the observations. Once the mating behavior of individuals was observed in the dark period, the mated female was replaced by another same-age virgin female in the coming light period, and the mated female was transferred into a new cage for further fecundity observations. Line the inner wall of the new cage with a piece of sulfuric acid paper, and replace the sulfuric acid paper with a new one daily at the onset of oviposition. The number of eggs and hatching eggs on the sulfuric acid paper were counted daily. After the death of the female moths, their abdomens were dissected to count the number of spermatophores in the spermathecae to determine mating frequency. The observation of mating behavior continued until the male died. A total of 24 males were observed.

### 2.3. Testing the Mating Potential of Female Beet Webworms

This operation process was similar to the experiment on determining male reproductive potential. The difference was that when mating behavior was observed in the dark period, the male mate would be transferred to a new cover (in the male group, the mated female would be removed) and replaced with another same-age virgin male in the following light period. Daily fecundity was checked and recorded during the light period. The mating behavior and fecundity observations were conducted daily until the female died. The abdomen of dead females was dissected, and the number of spermatophores was counted to correct the mating frequency after the female died. A total of 24 females were observed.

### 2.4. Effects of Sex Ratio on Reproduction and Longevity of Adults

According to our observation that the maximum mating frequency of females is about one in three males, we set a series of five sex ratios, ranging from a maximum of 3:1 to a minimum of 1:3, to test the effect of the sex ratio on reproduction and the population growth ratio. The newly emerged adults were paired according to the sex ratio series of 3:1 (9 females:3 males), 2:1 (8 females:4 males), 1:1 (6 females:6 males), 1:2 (4 females:8 males), and 1:3 (3 females:9 males) and raised in transparent plastic cylinder covers (8 cm in diameter, 22 cm in height). There were 12 adults in each replicate group and five replicates for each sex ratio treatment. Adults were fed a 10% glucose solution through cotton soaking. Eggs and hatching eggs were collected, and their numbers were recorded daily in each group. The adult mating frequency was also corrected by checking the number of spermatophores after a female died.

### 2.5. Reproductive Parameters

The maximum mating frequency, number of progenies, and progenies of the partners were used to assess the reproductive potential of each sex. The maximum mating frequency was the lifetime mating times of the male/female ratio on replacing female/male partners after each mating behavior occurred. The number of progenies in the male group was the sum of the lifetime fecundity of all the females, during which each of them mated once with the observed male, and that in the female group was the lifetime fecundity of the observed female that mated once with each of the male partners. The progeny of partners was the number of progeny divided by the number of opposite-gender partners (e.g., the number of once-mated female partners in the male group and the number of once-mated male partners in the female group).

Progeny per adult, progeny per female, progeny per male, and population net reproductive rate were used to assess the reproductive capacity under different sex ratios. According to the definition of net reproductive rate, which is the total mean number of offspring that an average individual (including females, males, and those who died in immature stages) can produce during their lifetime [36], we can use the net reproductive rate to assess the magnification of a population increase after one generation. In this study, we focus on the factors during the adult stage (sex allocation during the adult stage), and the factors during the immature stage were excluded; therefore, we use the calculation of progeny per adult to simplify that of the net reproductive rate. The parameters in the equation are: *R*_0_, net reproductive rate; *l_x_*, age-specific survival rate; *m_x_*, age-specific fecundity; *x* = age.
$$R_{0}=\sum_{x=0}^{\infty }l_{x}m_{x\;}=Progeny\;per\;adult=\frac{lifetime\;fecundity\;of\;all\;females}{number\;of\;both\;female\;and\;male}$$
$$Progeny\;per\;female=\frac{lifetime\;fecundity\;of\;all\;females}{number\;of\;female}$$
$$Progeny\;per\;male=\frac{lifetime\;fecundity\;of\;all\;females}{number\;of\;male}$$

### 2.6. Data Analysis

All numeric values are presented as the mean ± SEM. Independent-samples t test, Pearson’s correlation analysis, one-way analysis of variance (ANOVA), Tukey’s multiple comparisons tests, and quadratic curve regression analysis were used to determine significant differences (*p* < 0.05) between groups. Egg hatching rate data were arcsine transformed before statistical analysis. All statistical analyses were performed using SPSS software (SPSS 16.0).

## 3. Results

### 3.1. Mating Potential

The average maximum mating frequency of the male beet webworm was significantly higher than that of the female, which was 2.91 (4.95/1.70 = 2.91) times that of the female (Table 1, *t*_23,22_ = −6.42, *p* < 0.0001). Accordingly, the average number of maximum progenies of males was three times higher than that of females (*t*_23,22_ = −3.72, *p* = 0.001). At the same time, the net reproductive rate (*t*_23,22_ = −1.41, *p* = 0.17) between the female and male groups showed no significant difference between the sexes.

### 3.2. Effect of Maximum Mating Frequency on the Number of Progenies

The maximum mating frequency ranged between one and three times for the female beet webworm and one and 10 times for the male (Figure 1). With the increase in maximum mating frequency, the total number of progenies increased linearly in both the female and male groups (female, *F*_2,20_ = 6.21, *p* = 0.008; *y* = −8.84 + 102.52*x*, *r*^2^ = 0.82, *p* = 0.006; male, *F*_5,13_ = 3.38, *p* = 0.03; *y* = −126.15 + 120.00*x*, *r*^2^ = 0.97, *p* = 0.0002). However, the contribution to progeny at each mating decreased step by step, similar to a stair, in the male group with the increase in mating frequency (*F*_6,56_ = 3.51, *p* = 0.005; Pearson’s correlation analyses: *r* = −0.78, *p* = 0.04, *n* = 7; Figure 2).

### 3.3. Effect of the Sex Ratio on the Reproduction of Beet Webworms

The sex ratio significantly influenced the population's net reproductive rate (Figure 3, *F*_4,18_ = 9.22, *p* < 0.0001), progeny per male (*F*_4,18_ = 26.31, *p* < 0.0001), male mating frequency (*F*_4,18_ = 5.64, *p* = 0.004), and female mating frequency (*F*_4,18_ = 46.09, *p* < 0.0001) (Table 2). Among them, female mating frequency increased (Pearson’s correlation analyses: *r* = 0.70, *p* < 0.001, *n* = 23) and male mating frequency decreased (Pearson’s correlation analyses: *r* = −0.92, *p* < 0.001, *n* = 23), with a decrease in the sex ratio ranging from 3:1 to 1:3. In contrast, the sex ratio did not affect progeny per female (*F*_4,18_ = 2.78, *p* = 0.06) or egg hatching rate (*F*_4,18_ = 1.04, *p* = 0.42; Table 2).

Population’s net reproductive rate (*Y* = 90.57 + 62.68*x* − 14.39*x*^2^, *r*^2^ = 0.91, *p* = 0.04; Figure 3) fitted a parabolic curve with a decrease in female proportion ranging from a sex ratio of 3:1 to 1:3. Both of them under the sex ratio of 2:1 were significantly superior to those under 1:2 and 1:3 (*p* ≤ 0.012). Accordingly, the highest point indicated that the ideal optimal sex ratio for the population's net reproductive rate can be calculated and is at the sex ratio of 1.82, as shown using the parabolic curves (Figure 3). The progeny contribution per male decreased with a decreasing sex ratio from 3:1 to 1:3 (Pearson’s correlation analyses: *r* = −0.90, *p* < 0.001, *n* = 23). Progeny per male under sex ratios of 3:1 and 2:1 was significantly higher than that under the other sex ratios (*p* ≤ 0.001) (Table 2).

## 4. Discussion

### 4.1. Mating Potential and Multiple Mating in Beet Webworms

Our results support the previous conclusion that female and male beet webworms can multiply by mating with the same or different partners. Multiple mating in Lepidoptera and other taxa of insects is frequent. For instance, multiple mating was considered to serve as fertility assurance by increasing the production of offspring for the females of the red flour beetle (*Tribolium castaneum* Herbst) [26], the potato tuberworm *Phthorimaea operculella* (Zeller) [37], the female dobsonflies *Protohermes grandis* (Thunberg) [25], and the female seaweed fly *Coelopa frigida* [27].

Although female beet webworms may increase their fecundity by multiple mating, multiple mating is not simply the better way to encourage fertilization [38,39,40]. In the beet webworm, female fecundity increased with mating frequency when it was less than three times, while it decreased when mating was more than three times [41]. Although we did not test the mating frequency more than three times in the females in this study, it is reasonable to deduce that the contribution of the progeny number at each mating in females was not positively correlated with the mating frequency if it kept increasing and preceded a specific value. This was because mating frequency may trade off fecundity when the mating frequency overcomes a certain threshold. When the mating partner is unchangeable, female copulation times may be supplemented when mating is invalid, contributed by not transferring, or with very few fertile sperm by the male partner [26,42].

For males, multiple matings may increase the spread of sperm. However, the sperm transferred during each mating decreased with increasing mating frequency, which may lead to a decrease in the offspring obtained by a female partner during mating. In some lepidopterans, spermatophore size declined with the mating frequency of males [43,44]. Our results indicate that females and males could benefit from multiple matings by passing on reproductive resources as much as possible to the next generation. However, maximum mating may trade off the reproductive resources of the population, involving a decrease in the effective sperm to transfer to females. This situation decreases the opportunity for a virgin female to mate with a virgin male, which may transfer more effective sperm through mating. Consequently, maximum mating leads to a decrease in the benefit of the demographic condition.

If the maximum mating frequencies were the ultimate judging factor, then the best sex ratio (under the best sex ratio, the population should have the best net reproductive rate) between females and males should be 2.91 (4.95/1.70 = 2.91); in the meantime, the offspring of males should be 2.91 times that of females; however, our results denied this assumption. Because the population’s net reproductive rates had no difference between the female and male groups when females or males released their maximum mating potential alone, this indicated that the sex ratio of the population with the maximum net reproductive rate is not the same as that under which both males and females, or either of them, can release their maximal mating potential. If the assumed female and male reproductive resources are considered the restrictive factor in a population, then the population’s net reproductive rate should be a reliable indicator that reflects whether both sexes have released their reproductive potential by presenting its value. If a population presents the maximum value of the population’s net reproductive rate, it should proportion the sex ratio to be under what both sexes can release their reproductive potential; however, not the maximum mating frequency that may be achieved.

### 4.2. Sex Ratio and Population Outbreaks

Our results showed that the population's net reproductive rate peaked at a sex ratio of 1.82, according to the prediction of the curve regression model. This result indicates that the closer the 1.82 sex ratio is, the faster the net reproductive rate should be in beet webworms. These results explain why the year when the first generation of larval beet webworm outbreak always accompanied the overwinter generation adult sex ratio was near two by the observation of historical data in Kangbao County. In contrast, the year when the population broke down was usually concurrent with the parent generation, with an adult sex ratio lower than one.

As a typical lepidopteran, the ratio of females to males should be 1:1 in the next generation of heredity. The sex ratio fluctuated around 1:1 in the laboratory population of beet webworms [12] and most of the occurrence years in the field, except for the occurrence peak year, were near to 2:1. Since sex ratio variation is closely connected with population dynamics, it would be reasonable to deduce that the sex ratio can be biased by population density or other biological factors. The beet webworm is a typical migratory insect population. The population usually decreases from the blooming point in the next year, even when it is still in the occurrence cycle of a period. Our previous laboratory research results [45] showed that mortality is higher when rearing density is pressured to 30 larvae per jar (650 mL) or more compared to the average high density of 10 larvae per jar, which indicated that the mortality of individuals in the population increased with the increase in food and space pressure. If the mortality of females and males is similar under density stress, then the sex ratio should not be biased by the increasing population density. In contrast, the resistance to female and male density pressure differed in our previous study [46]. It may be a general phenomenon in the periodic outbreak of insects that male mortality is higher than female mortality at low population densities. In comparison, female mortality is higher than male mortality when the population density is high [19].

During evolution, reproduction output in females was higher than that of males for the breeding of offspring. As a trade-off, some somatic production in females may be lower than that in males, such as resistance to ambient stress, including food, space, pathogeny, or the superiority of interpopulation competition [47,48,49]. In addition to the resistance difference, males may allocate more resources to moving, activities involving activity, and mating competition [22]. Males may show high fitness in a high-density life, requiring more exercise than females. When the population density is lower, males may not need to move or move far to search for food. The decrease in movement may not be compatible with the male body mechanism, in which most of the locomotive-correlated metabolic genes would not be activated [50]. In turn, this results in male resistance not arriving at an average level and causing more mortality in males than in females in the lower population density [46]. In contrast, at a high density, males may need more movement and competition for food and space, inspiring relative neuropeptide or neuroamine synthesis and contributing to the low mortality of males in the face of intrapopulation competition [50]. In contrast, females may be more suitable for loose habitats for offspring breeding and show a high survival rate under a lower population density than males [51].

Accordingly, we inferred that it was through influencing the sex ratio that population density could contribute to regulating the population dynamic itself and hence move the occurrence period cycle of the population. That was when the sex ratio decreased from approximately two at the peak point year to less than one at the bottom occurrence year when the population was broken down. High-density stress may cause more female deaths than male deaths during an immature stage, while low-density stress is available to increase the number of females. Moving by this effect, the sex ratio structure may be shaped from the highest of two to the bottom of less than one, then from the bottom to the top again. This process goes back and forth and is repeated in the occurrence dynamics.

In response to the variation in the sex ratio, the beet webworm outbreaks in the year of the sex ratio peak point indicated the onset of a new occurrence period. With the decline in the sex ratio, the population decreased to approximately one, and the population entered an average occurrence period. When it declined to less than one, it marked the end of an occurrence period, and the population entered the next interoccurrence period. In the interoccurrence period, the sex ratio repeats the former recovery process of increasing annually until the next occurrence period. The interoccurrence and occurrence periods were altered, and the occurrence cycle period was repeated every 18–25 years in the beet webworm [34].

It is worth mentioning that although population density can facilitate self-regulation through sex ratio, sex ratio is not an independent factor. The regulation of the sex ratio itself is influenced by various factors. In addition to population density, food, natural enemies, and even climate factors can also have an impact on the sex ratio [19], or the sex ratio itself is also a regulatory result of environmental factors. In addition to the sex ratio, environmental factors can even directly affect population density to regulate population dynamics. It is precisely because of the complexity of population dynamics and regulation that an accurate forecast has not been achieved since the first records of the occurrence of beet webworms began in China. Determining the end and beginning of a periodical cycle based on a definite indicator is the key to solving this problem. The connection between the sex ratio and the explicit value of the sex ratio (1.82) in this study may provide a definite indicator to depend on for the forecast of population dynamics and, in turn, for the theoretical study of population in the beet webworm.

## Figures and Tables

**Figure 1 insects-14-00781-f001:**
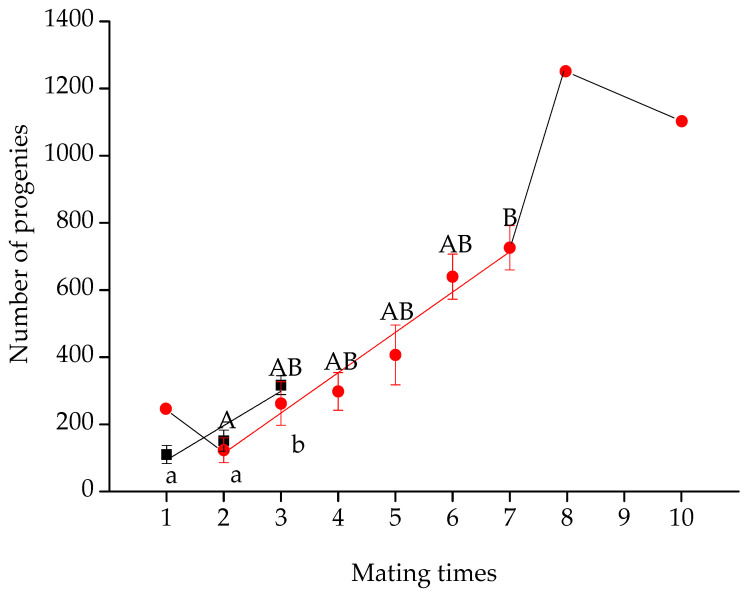
Number of progenies affected by mating times in male and female beet webworms. Lines show data-fit linear regression analysis (black line shows female, *y* = 9.41 + 85.99*x*, *r*^2^ = 0.27, *p* = 0.006; red line shows male, *y* = −72.72 + 107.73*x*, *r*^2^ = 0.41, *p* = 0.0007). Lowercase letters indicate significant differences between the female groups. In contrast, uppercase letters show males. For males, it represents the number of progenies of all female partners mated with them. Since treatments at male mating times of 1, 8, and 10 had no replicates, data from these groups was excluded from the statistical analysis.

**Figure 2 insects-14-00781-f002:**
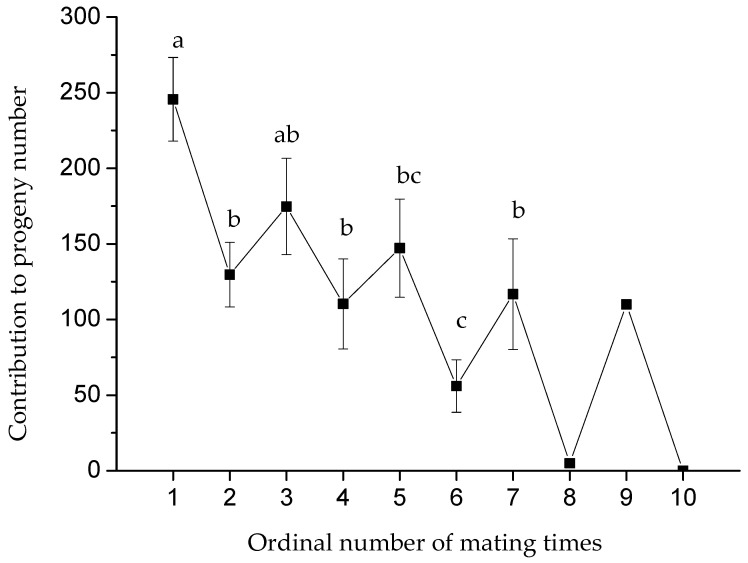
Contribution to progeny number per mating is affected by the ordinal number of mating times in male beet webworms. Lowercase letters indicate the significant difference among the groups of ordinal mating times. Since treatments at male mating times of 8, 9, and 10 had no replicates, data from these groups were excluded from the statistical analysis.

**Figure 3 insects-14-00781-f003:**
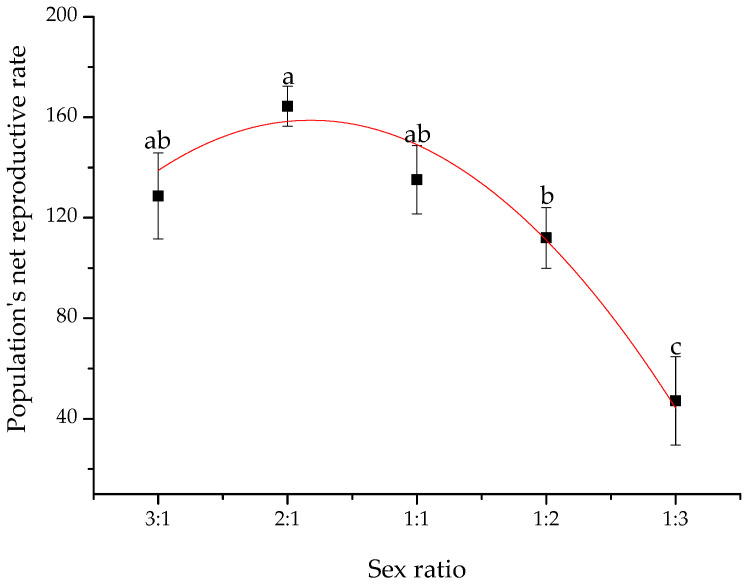
The population's net reproductive rate is affected by the sex ratio in the beet webworm. The population’s net reproductive rate was calculated, simplified by progeny per adult. Plots sharing the same letter indicate no significant difference between the treatments. The red line shows that the parabolic curve regression analysis was significant (*y* = 90.57 + 62.68*x* − 14.39*x*^2^, *r*^2^ = 0.91, *p* = 0.04).

**Table 1 insects-14-00781-t001:** Mating potential and mating potential dependent reproduction in the beet webworms.

Sex	Maximum Mating Frequency	Number of Progenies	Net Reproductive Rate
Female	1.70 ± 0.15 *	154.96 ± 22.57 *	56.18 ± 7.49
Male	4.95 ± 0.49	461.05 ± 79.05	73.59 ± 9.92

For males, it represents the number of progenies of all female partners mated with them. The asterisk (*) indicates a significant difference between females and males tested by using an independent-sample *t* test at the 0.05 level. The sample numbers for the female and male groups were 23 and 22, respectively.

**Table 2 insects-14-00781-t002:** Effect of the sex ratio on reproduction of beet webworms.

Sex Ratio (Female vs. Male)	Progeny per Female	Progeny per Male	Hatching Rate (%)	Female Mating Frequency	Male Mating Frequency
3:1	171.62 ± 22.81 a	514.86 ± 68.44 a	83.40 ± 1.17 a	1.00 ± 0.10 c	3.00 ± 0.30 a
2:1	246.65 ± 12.00 a	493.30 ± 24.00 a	84.80 ± 1.93 a	1.23 ± 0.05 bc	2.45 ± 0.09 a
1:1	270.29 ± 27.12 a	270.29 ± 27.12 b	85.00 ± 1.78 a	1.29 ± 0.14 b	1.29 ± 0.14 b
1:2	315.85 ± 37.16 a	157.92 ± 18.73 bc	86.60 ± 1.03 a	1.3 ± 0.05 b	0.65 ± 0.03 bc
1:3	183.36 ± 71.42 a	61.12 ± 23.80 c	88.00 ± 2.68 a	1.58 ± 0.08 a	0.56 ± 0.00 c

Different lowercase letters indicate the significant difference between the groups of sex ratios. The sample numbers in each column from the top list are listed as 5, 5, 4, 5, and 4, respectively.

## Data Availability

Data are available upon request from Xingfu Jiang.

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
