# Peer review of "The Sex Ratio Indicates the Conclusion and Onset of Population Cycles in the Beet Webworm Loxostege sticticalis L. (Lepidoptera: Pyralidae)"

_insects, 2023, doi:10.3390/insects14100781_

Round 1
Reviewer 1 Report
Comments to the Authors
This submitted paper explains the topic of sex ratio and population control using the beet webworm as a case study from a theoretical perspective. Many leaps and errors in interpretation (my opinion) have been discovered, making it difficult to accept it as is and requiring sufficient revision. However, it seems to be worth it because it contains very progressive and challenging content that can lead to the development of this field.
I expect the authors to respond fully to my comments. Especially, comments for Discussions.
Line 42. The sex ratio is one of the most critical biological characteristics influencing the growth rate of the arthropod population. And Line 165 population growth rate.
---> It is necessary to mention the life table statistics. In other words, the intrinsic rate of natural increase is the key parameter for the population growth model.
It is difficult to interpret what does mean the concept of population growth rate in Line 165.
The concept of Line 165 as a population growth rate is thought to be meaningless. Please provide suitable citations to justify this concept.
Therefore, Figure 4 is also meaningless because it is not differentiated from Figure 3.
To present both Figures, it is necessary to justify what the progeny per adult and population growth rate mean ecologically or conceptually.
Line 45. This happens especially for migratory insect species, where population outbreak years are often at a certain sex ratio.
---> Require references
Line 83 To 98
What is currently described in Appendix Fig. A1 is difficult for me to understand.
Comments or questions for Fig. A1.
First, the generation larval damage area in Kangbao County found that the sex ratio was not stabilized at the ideal value of 1:1 as a typical Lepidoptera species.
--->The sex ratio of insects can vary locally. In order for the sex ratio shown in the figure to be valid, it is necessary to ensure that there is a sufficient sample size. Indicating the sample size and number of repetitions would provide more confidence.
Nevertheless, it varied from 0.7 to 2.2, and the first generation of larval damage area was strongly positively correlated with the sex ratio in this range.
---> At first glance, there is not a clear correlation between sex ratio and damage area. Rather, it seems to be related to the number of overwintered adults and damage area. There is a need to clarify the concept of correlation.
In addition, during the damage period of beet webworms, when the sex ratio was lower than one, regardless of the damaged area of the first-generation larval population, the first-generation larval damage area will be very low.
---> These sentences are difficult to understand because they contradict each other (regardless of the damaged area of the first-generation larval population vs. the first-generation larval damage area).
This is called population collapse and indicates the end of a fluctuation period. Later, the population enters an inter-occurrence period, during which the larval number and larval damage area are shallow, which lasts for years
---> the larval number ---> overwintered adult number ?
For example, does the inter-occurrence phase refer to the period from 1986 to 2009? Or from 1986 to 1996? Please specify approximately the period of the interphase.
This period would not end until the sex ratio recovered from a bottom point of less than one to the next nearest top point of more than two.
---> So the top point would be around 1987? Please specify the year around the top sex ratio.
Then, the population entered another occurrence period. During a new occurrence period, the sex ratio fluctuated from one to two. This period usually continued for 10–12 years until the value decreased to another low point lower than one and recovered again. This process repeated itself, and the beet webworm population broke out periodically.
---> Approximately from what year will another occurrence period begin?
The sex ratio fluctuated from one to two in most years except of 1986 and 2009.
Line 111. a laboratory population
---> Please provide the collected year of source population.
Line 123. the mated female was replaced by another virgin female in the following light period, and the mated female was transferred into a 124 new cage for further fecundity observations.
---> Please provide oviposition substrate
The longevity of males should be provided in a Table. Maximum mating times of males was 10 times, then this case male longevity will be maximum 40 d because the first mating of females was about 4 d.
Additionally, further explanation is needed as to what condition the female was placed in. Was it a newly emerged adult or a 4 day old female? Was the experiment designed so that females could mate right away?
Further, the condition of males should be described in the experiment of Testing the mating potential of female beet webworms.
Line 126. After the death of the female moths, their abdomens were dissected to count the number of spermatophores in the spermathecae to determine mating frequency; Line 147. The adult mating frequency was also corrected by checking the number of spermatophores after a female died.
---> Not spermathecae, probably bursa copulatrix
Spermatophores can be broken down and become a source of nutrition for the female sometimes in Lepidoptera. It must be possible to prove that spermatophores can exist after the female's death. A detailed explanation is needed on how the spermatophores were investigated. It is necessary to provide supplementary materials such as photos.
Line 188. Table 1
Does “the average number of maximum progenies of males (i.e., 461.05)” indicate the number of offsprings of all female partners mated with?
Need a footnote below the Table.
Line 196. (female, F 2, 20 = 1.25, P = 0.008; y = 9.41 + 85.99x, r2 = 0.27, P = 0.006; male, F 5, 13 = 3.38, P = 0.03; y = −72.72 + 107.73x, r2 = 0.41, P = 0.0007).
--->Visually, Figure 1 appears to fit the data well (i.e., linear line), but isn't the coefficient of determination too low? Please check the statistics.
Please check of d.f. (5, 13) for males; total 22 males, then d.f. should be 5, 16.
Please check the significant level for “F 2, 20 = 1.25, P = 0.008”, here P=0.008 is not possible because of F=3.49 at 5% level with d.f.=2, 20
Line 202. Figure 1.
Why no data point at 9 mating times, but it is seen in Figure 2.
Please include a footnote as “For males, it represents the number of progenies of all female partners mated with”
Line 246. Discussion
The authors considered that the population dynamics of this pest could be entirely controlled by the sex ratio. It is clear that there is variation in sex ratios in field populations. And I agree with that. I also agree that laboratory studies show differences in the number of offspring according to sex ratio.
The authors need to examine that sex ratio may occur as a result of population size fluctuations rather than as a cause of population dynamics. In the basic population growth model, the population size decreases only when the intrinsic rate of natural increase (rm) is less than 0. Can we say that when the sex ratio becomes less than 1, the population growth rate (rm) becomes less than 0?
I believe that the long-term population dynamics of this pest, as presented in the Appendix Fig. A1, are not controlled by sex ratio but are the result of various other density regulating factors, i.e., density-dependent and independent factors. Various biotic an abiotic factors such as diseases, natural enemies, precipitation, etc.
Various factors may have played a role in controlling the population dynamics of this pest. I hope the authors find these factors first. Again, the sex ratio may be a result of other factors. If sex ratio were the main factor, it would be difficult for phenomena such as the inter-occurrence period to appear. If the sex ratio is recovered, the population size should also be recovered immediately. As shown in the Fig. A1, the reason why the population continues to show an inter-occurrence period after the recovery in the sex ratio is because various other factors are at work.
Reviewer 2 Report
Please find my revision in the attached file.

Some of the sentences require rephrasing either because of their length or because of their context.
Round 2
Reviewer 1 Report
We believe that this manuscript has been greatly improved after revision. The authors responded point by point to the comments.
I understand the authors tried to define the ending or the beginning of a population damage cycle in the beet webworm by the sex ratio. Anyway, one thing I ask the authors to add to the discussion (see response 14) is the statement that sex ratio may not be the cause of population regulation, but results. The current status may mislead readers into thinking that sex ratio is a key factor in population control.
Author Response
Comments 1:
We believe that this manuscript has been greatly improved after revision. The authors responded point by point to the comments.
I understand the authors tried to define the ending or the beginning of a population damage cycle in the beet webworm by the sex ratio. Anyway, one thing I ask the authors to add to the discussion (see response 14) is the statement that sex ratio may not be the cause of population regulation, but results. The current status may mislead readers into thinking that sex ratio is a key factor in population control.
Response 1
We really appreciated for this valuable suggestion. We checked the discussion seriously to make sure that the description about sex ratio is an indicator but not the cause of the population dynamics. These are three objectives: population density-sex ratio-population dynamics, population density is the cause, sex ratio the pathway, population dynamics the results. If we cut the connection between sex ratio and population dynamics, then this article can not be built. So we kept the previous version of statement in the Part 4.2 Sex ratio and population outbreaks. And we added a paragraph of statement to declare the relationship between the three objectives.
Line 363-374
“It is worth mentioning that although population density can realize self-regulation through sex ratio, sex ratio is not an independent factor. The regulation of sex ratio itself is influenced by various factors. In addition to population density, food, natural enemies, and even climate factors can also have an impact on sex ratio [19], or sex ratio itself is also a regulatory result of environmental factors. In addition to through sex ratio, environmental factors can even directly affect population density to regulate population dynamics. It is precisely because of the complexity of population dynamics regulation that an accurate forecast has not been achieved since the first records of the occurrence of beet webworms began in China”.
Hope this will work. Thank you very much!